# Effect of $^2$H and $^{18}$O water isotopes in kinesin-1 gliding assay

Andy Maloney[1], Lawrence J. Herskowitz[2] and Steven J. Koch[3]

[1] College of Pharmacy, The University of Texas at Austin, Austin, TX, USA
[2] Fann Instrument Company, Houston, TX, USA
[3] College of University Libraries & Learning Sciences, The University of New Mexico, Albuquerque, NM, USA

## ABSTRACT

We show for the first time the effects of heavy-hydrogen water ($^2$H$_2$O) and heavy-oxygen water (H$_2$$^{18}$O) on the gliding speed of microtubules on kinesin-1 coated surfaces. Increased fractions of isotopic waters used in the motility solution decreased the gliding speed of microtubules by a maximum of 21% for heavy-hydrogen and 5% for heavy-oxygen water. We also show that gliding microtubule speed returns to its original speed after being treated with heavy-hydrogen water. We discuss possible interpretations of these results and the importance for future studies of water effects on kinesin and microtubules. We also discuss the implication for using heavy waters in biomolecular devices incorporating molecular motors.

## INTRODUCTION

Water plays a crucial role in the interactions of biomolecules especially when biological surfaces bind and unbind. When binding surfaces are apart, water molecules form a hydration shell around the surfaces that is more ordered and less dynamic than bulk water molecules (*Israelachvili & Wennerström, 1996*). Changes in osmotic stress affect the rate of transport of water between the hydration shells around surfaces and the bulk water (*Parsegian, Rand & Rau, 2000*). Thus, on- and off-rates for surface-surface binding is strongly affected by changes in the activity of bulk water. As noted by Parsegian, water activity changes that affect on- and off-rates are often overlooked, even in meticulous biophysical studies (*Parsegian, Rand & Rau, 1995*). In fact, water is an often-overlooked yet vitally important aspect for biophysical systems and is "the most important problem in science that hardly anyone wants to see solved" (*Cho, Singh & Robinson, 1996*). A good example of the profound importance of water activity is shown by a more than three orders of magnitude increase in binding lifetime in a protein-DNA complex when the osmotic stress of the system is increased by the addition of betaine (*Sidorova & Rau, 2001*). The effects of water activity have not been extensively studied in the molecular motors field, and in particular, for the kinesin-1 and microtubule system. In this study, we show the effects of changing the water isotope has on the speed of gliding microtubules using kinesin-1.

Corresponding author
Andy Maloney,
amaloney@austin.utexas.edu

Though understudied for molecular motors, in general there have been some experiments indicating the importance of water on actin and myosin. For example, *Highsmith et al. (1996)* performed a detailed study on the effects of osmotic stress on the activity of myosin and binding to actin. They demonstrated that changing osmotic pressure was effective for probing water molecules at the binding interface. The authors also point out the importance of the increased osmotic pressure inside living cells. *Chaen et al. (2001; 2003)* also showed a 60% reduction in *in vitro* actin gliding velocity in heavy-hydrogen water.

We are unaware of prior studies involving use of water isotopes as osmotic stressors using the kinesin and microtubule system. However, there have been several studies of osmotic stress effects on tubulin alone and on microtubule polymerization. It is well known that osmotic stress promotes microtubule polymerization and a high percentage of glycerol is often used to promote *in vitro* microtubule polymerization (*Shelanski, Gaskint & Cantort, 1973*). It has also been shown that heavy-hydrogen water promotes tubulin polymerization and stabilization (*Houston et al., 1974*; *Panda et al., 2000*). Heavy-hydrogen water stabilizes polymerized microtubules, which has been shown to be a likely reason for why deuterium oxide is highly toxic to eukaryotic organisms (*Kushner, Baker & Dunstall, 1999*). *Lewis* (*1933*; *1934*) initially demonstrated the toxicity of deuterium oxide in the 1930s. The ability of tubulin to polymerize fades rapidly when stored in normal aqueous buffer at 4 °C, however, heavy-hydrogen water dramatically reduces this instability allowing tubulin to polymerize even after storage at 4 °C for 2 days (*Chakrabarti et al., 1999*). Heavy-hydrogen water has also been shown to stabilize a wide range of biomolecules and has potential technological implications for vaccine stability (*Sen et al., 2009*). It is likely that heavy-hydrogen water has a general stabilizing effect on biomolecules and biomolecular complexes as it has also been shown to protect fruit flies from elevated temperature (*Pittendrigh & Cosbey, 1974*).

Despite its importance, it can be argued that water solvent effects have seen relatively little attention in the kinesin and microtubule system. Water isotopes have been used, but usually as spins or tracers rather than a probe of water activity (*Hackney, Stempel & Boyer, 1980*). In this paper, we show that heavy-hydrogen water and heavy-oxygen water each slow down microtubule gliding speeds on kinesin-1 surfaces. Thus, water isotopes may provide an important experimental knob to turn while studying the effects of water interactions on the behavior of kinesin-1 in cells and biomolecular devices.

## MATERIALS AND METHODS

### Microscope

Experiments were conducted on an Olympus IX71 inverted microscope using an Olympus 60x 1.42 NA PlanApo objective. Rhodamine fluorophores attached to tubulin were illuminated using a 100 W mercury lamp and excited using a TRITC filter cube from Chroma (Chroma 49005). The light illuminating the fluorophores was attenuated by 94% in order to prevent excessive photobleaching. The objective was temperature stabilized to achieve consistent microtubule speed measurements. A custom objective heater was

constructed for the temperature stabilization and for a more in depth description of the heater please see *Maloney, Herskowitz & Koch (2011)*. Briefly, the objective heater used a polyimide film resistive heater to heat the objective using control circuitry from TeTech (TC-48-20). LabVIEW software supplied by TeTech was used and modified to suit our time-stamping requirements. Temperature stabilization plays a crucial role in obtaining stable measurements (*Highsmith, 1977*; *Hackney, Stempel & Boyer, 1980*; *Böhm et al., 2000*; *Böhm, Stracke & Unger, 2000*); however, many studies do not indicate the temperature at which speed measurements were observed. All our experiments were conducted at $33.1 \pm 0.1\,°C$. Image sequences were captured every 200 ms for a total of 600 frames—approximately 2 min—with an Andor Luca S EMCCD camera. Images were stored in PNG format using custom LabVIEW software.

## Flow cells

Flow cells were constructed using two strips of double stick tape (Scotch), sandwiched between a microscope slide (VWR 48300-025) and a cover slip (VWR 48366-045). The channel formed between the pieces of tape was approximately 10 μL in volume and was sealed using nail polish to prevented excessive evaporation of the motility solution during observations. In order to determine if heavy-hydrogen water affected speed measurements, we also created flow cells that could be "sealed" and "unsealed" with cellophane. A complete description of our procedure can be found in Text S1. Briefly, resealable flow cells were constructed in the same manner as sealable ones with the addition of two very thin pieces of double stick tape placed on top of the slip and at the entrances of the channel. Cellophane (Glad Cling Wrap) was then wrapped around the entrances of the flow cell in order to prevent the motility solution from evaporating.

## Buffers and solutions

We used four different solvents to make the buffer solutions used in our experiments; light water ($H_2O$), heavy-hydrogen water ($^2H_2O$), heavy-oxygen water ($H_2{}^{18}O$), and deuterium-depleted light water.

The light water buffer was made at a 10x concentrated solution and contained: 800 mM PIPES (Sigma 80635), 10 mM EGTA (Sigma 80635), 10 mM $MgCl_2$ (Sigma M1028) and pH-ed to 6.89 using approximately 1.25 M NaOH (Fisher S318). The light water used for this buffer had a resistivity of 18.2 MΩ-cm and was produced with a Barnstead EasyPure RoDI system. The light water buffer was passed through a 0.2 μm syringe filter and aliquoted in 2 mL screw-top vials, stored at 4 °C, and stored for no longer than 6 months.

Another 10x concentrated buffer solution containing the same 800 mM PIPES, 10 mM EGTA, and 10 mM $MgCl_2$ was prepared using heavy-hydrogen water (Sigma 151882) and pH-ed to a value of 7.30 using NaOH. In order to report the correct pD value of a solution containing heavy-hydrogen water, 0.41 must be added to the measured value of pH (*Covington et al., 1968*). Thus our solution had a pD value of 7.71, which is higher than the pH value for the light water buffer. The difference in the pH values of these two buffers and the consequence of those differences in the measured speed value is discussed below.

The heavy-hydrogen water buffer was syringe filtered using a 0.2 μm filter and aliquoted in 2 mL screw-top vials, stored at 4 °C, and stored for no longer than 6 months.

The third buffer solution contained heavy-oxygen water (Sigma 329878). A 90% dilution of the 10x light water buffer was added to the heavy-oxygen water in order to create a solution containing 80 mM PIPES, 1 mM EGTA, and 1 mM $MgCl_2$, which was the motility buffer conditions. We diluted the light water buffer into the heavy-oxygen water due to the high cost of the heavy-oxygen water. The solution was not filtered for fear of losing too much material and stored at 4 °C for no longer than 6 months.

The fourth buffer again was a 10x concentrated solution and contained 800 mM PIPES, 10 mM EGTA, 10 mM anhydrous $MgCl_2$ (Sigma 449172) and pH-ed to 6.89 with NaOH using deuterium-depleted light water as the solvent (Sigma 195294). The deuterium-depleted light water was quoted to have less than 1 ppm deuterium oxide. The deuterium-depleted light water buffer was made with anhydrous $MgCl_2$ in order to reduce the amount of contaminant deuterium in solution. It was syringe filtered using a 0.2 μm filter and aliquoted into 2 mL screw-top vials, stored at 4 °C for no longer than 6 months.

1x concentrated solutions of the light water buffer, heavy-hydrogen water buffer, and deuterium-depleted light water buffer were made with the addition of 1 mg/mL bovine $\alpha$-casein (Sigma C6780) in solution in order to passivate the glass slide for the gliding motility assay. Bovine $\alpha$-casein was chosen as the surface passivator as it has been shown to give consistent gliding motility assays (*Maloney, Herskowitz & Koch, 2011*), however, it is not the only passivator that can be used to conduct gliding motility assays (*Miller-Jaster, Petrie Aronin & Guilford, 2012*). No $\alpha$-casein solution using the heavy-oxygen water buffer was made due to the cost of the $H_2{}^{18}O$ water. The light water buffer containing $\alpha$-casein was used for the surface passivation when using heavy-oxygen water buffers. All $\alpha$-casein solutions were mixed for approximately one hour at room temperature using a stir plate until no visible precipitates of $\alpha$-casein were left in solution. These solutions were not filtered and were aliquoted into 2 mL screw-top vials and stored at 4 °C. No investigations were made to determine the solubility of $\alpha$-casein in the various water buffer solutions.

Tubulin was polymerized into microtubules using a Thermo PCR Sprint thermal cycler held at a constant temperature of 37 °C for 30 min. The tubulin used for polymerization consisted of 29% rhodamine-labeled bovine tubulin (Cytoskeleton TL331M) and 71% unlabeled bovine tubulin (Cytoskeleton TL238) at a concentration of 5 mg/mL. The polymerization solution was a total volume of 1 μL and contained: light water buffer, an extra 1 mM $MgCl_2$, 1 mM GTP (Sigma G8877), 6% (v/v) glycerol (EMD GX0185) and tubulin. Adding an extra 1 mM $MgCl_2$ to the light water buffer ensured the EGTA did not chelate all the magnesium ions from solution since tubulin polymerization requires magnesium ions (*Olmstedt & Borisy, 1975*). Tubulin will polymerize in the presence of ATP; however, GTP is known to be a better nucleotide for polymerization (*Desai & Mitchison, 1997*). Glycerol acts as an osmotic stress in the polymerization solution and helps to speed up the polymerization process (*Shelanski, Gaskint & Cantort, 1973*). Microtubules were formed after 30 min in the thermal cycler and diluted by 200x with a solution of 10 μM Taxol[TM] (Cytoskeleton TXD01) in light water buffer. Taxol[TM] is a

non-polar chemical that must first be dissolved in DMSO (Sigma D2650) before adding to an aqueous solution. Taxol$^{TM}$ has been shown to create crystals in aqueous solutions due to its low solubility that appear to be fluorescent microtubules (*Foss et al., 2008*). It also has a high affinity for free tubulin and/or rhodamine dye molecules (*Castro et al., 2009*; *Castro et al., 2010*). In order to prevent Taxol$^{TM}$ crystals from forming in solution, all solutions requiring Taxol$^{TM}$ were prepared immediately before each experiment and no stock solutions containing Taxol$^{TM}$ in an aqueous environment were stored for more than the duration of daily experiments. Microtubules were left at room temperature for the duration of an experiment in order to prevent depolymerization (*Shelanski, Gaskint & Cantort, 1973*).

Dr. Haiqing Liu generously supplied Kinesin to us in 20 μL aliquots at a concentration of 0.275 mg/mL kinesin. The kinesin was his-tagged, truncated kinesin-1 dmk401 (*Berliner et al., 1995*; *Asbury, Fehr & Block, 2003*) from drosophila and was expressed in *E. coli*. Kinesin was diluted to 27.5 μg/mL for each assay.

## Motility assays

All motility assays contained the following components; 10 μM Taxol$^{TM}$, 1 mM Mg-ATP (Sigma A9187), 20 mM D-glucose (Sigma 49139), 2.5% (v/v) of an oxygen scavenging antifade cocktail, and 5 mL of fixed polymerized microtubules. The antifade cocktail was a dual enzymatic oxygen scavenging system, which consisted of; 800 mg/mL glucose oxidase (Sigma G6641), 2000 mg/mL catalase (Sigma C9322) and 20% (v/v) of 2-mercaptoethanol, BME, (Sigma 63689). When diluted into the motility solution, there was 8 mg/mL glucose oxidase, 20 mg/mL catalase and 0.5% (v/v) BME. Antifade cocktails were prepared in advance and stored in 5 mL aliquots at −20 °C and used within one week.

Experiments using light water required the flow cell to be passivated with 1 mg/mL α-casein in the 1x light water buffer for 10 min. Similarly, experiments using heavy-hydrogen water were incubated with 1 mg/mL α-casein in the 1x heavy-hydrogen water buffer, and those experiments using deuterium-depleted light water used 1 mg/mL α-casein in the 1x deuterium-depleted light water buffer. 1 mg/mL α-casein in the light water buffer was used as the passivator in the heavy-oxygen water buffer experiments due to cost.

Each experiment incubated the flow cells at room temperature, 24 °C, for 10 min. During this 10-min incubation, kinesin was diluted to 27.5 μg/mL in a 20 μL solution of either 1x light water buffer, 1x heavy-hydrogen water buffer, heavy-oxygen water buffer, or 1x deuterium-depleted light water with the addition of 1 mM Mg-ATP and 0.5 mg/mL α-casein. After the 10-min incubation with α-casein, the 20 μL kinesin solution was introduced to the flow cell by fluid exchange. The flow cell was then allowed to incubate for another 5 min. During this second incubation time, a motility solution was prepared using the different water isotope buffers. After the 5-min incubation, 20 μL of the motility solution was flown into the flow cell by fluid exchange. The flow cell was then sealed and observations were conducted immediately on the microscope. All observations were done in the center of the flow cell far from the tape and nail polish boundaries since this region gave the most consistent gliding motility speeds (*Maloney, Herskowitz & Koch, 2011*).
## Experiment and data collection

Images were captured every 200 ms. A total of 600 frames—approximately 2 min—were captured for a given region of interest (ROI) using an EMCCD gain of 150 with an Andor Luca S camera. The use of the antifade cocktail, attenuation of the mercury lamp, and proper Köhler illumination helped prevent excessive photobleaching of the sample. A total of 15 ROIs were taken for each slide prepared. Due to the incubation of the flow cells at room temperature, it took approximately five 2 min intervals until the slide reached a stable temperature set by the objective heater from which consistent speed measurements were achieved. Each water condition was repeated for a total of 3 independent flow cells with 15 ROIs taken for each sample.

Images were analyzed using custom LabVIEW 7.1 software to track and report the speed of each microtubule in an ROI. NI Vision 7.1 image segmentation algorithms via pattern matching identified microtubules. Tracking of microtubules was stopped if the microtubule end was within a few pixels from the boarder of the ROI or if a microtubule overlapped with another microtubule. If a microtubule had fewer than 100 consecutive image frames without tracking problems, the track was discarded. Microtubules that had a segmented area less than 55 pixels were also discarded. These values were determined empirically and were found to provide well-tracked microtubules with few tracking errors.

Tracking provided the $x$ and $y$ position of microtubule ends with sub-pixel accuracy. This time series data was smoothed using a Gaussian window with a standard deviation of 2 s in order to eliminate transverse Brownian noise. Figure 1 shows a typical position trace with speed data from a microtubule using the light water buffer. The markers in Fig. 1A are of the raw speed data obtained from the raw position data in Fig. 1B, also shown as markers. The smoothed position data in Fig. 1B (solid magenta line) is then used to calculate the smoothed speed data of the microtubule in Fig. 1A (solid magenta line). Smoothed data within 5 s of the beginning and end of a microtubule track were discarded to eliminate edge effects due to the Gaussian smoothing and are shown as the black solid lines in Fig. 1A. The solid magenta line in Fig. 1A was the data used to calculate speed values for microtubules.

Speed versus time data for all the microtubules in an individual ROI were then concatenated together and the most likely speed was extracted using a kernel density estimation (KDE) using a Gaussian kernel of width 50 nm/s (*Silverman, 1986*). Using a KDE method instead of determining a simple mean speed reduced our sensitivity to microtubule pausing, stalling, or tracking errors. Figure 2 shows smoothed speed variations when using different water isotopes in the motility solution. Figure 2A shows truncated smoothed speed data concatenated together for the heavy-hydrogen water buffer. It is clear that the microtubules do not travel at a single speed and can go through periodic pausing or stalling events. As greater amounts of the different water isotopes were added to the motility solution, microtubules began slowing and more of them had more instances of stalling or pausing. Figure 2B shows concatenated sorted smoothed speed values for the heavy-hydrogen and heavy-oxygen water buffers as compared to a light water buffer (solid black line). Sorting the speed values clearly shows the effects of the water
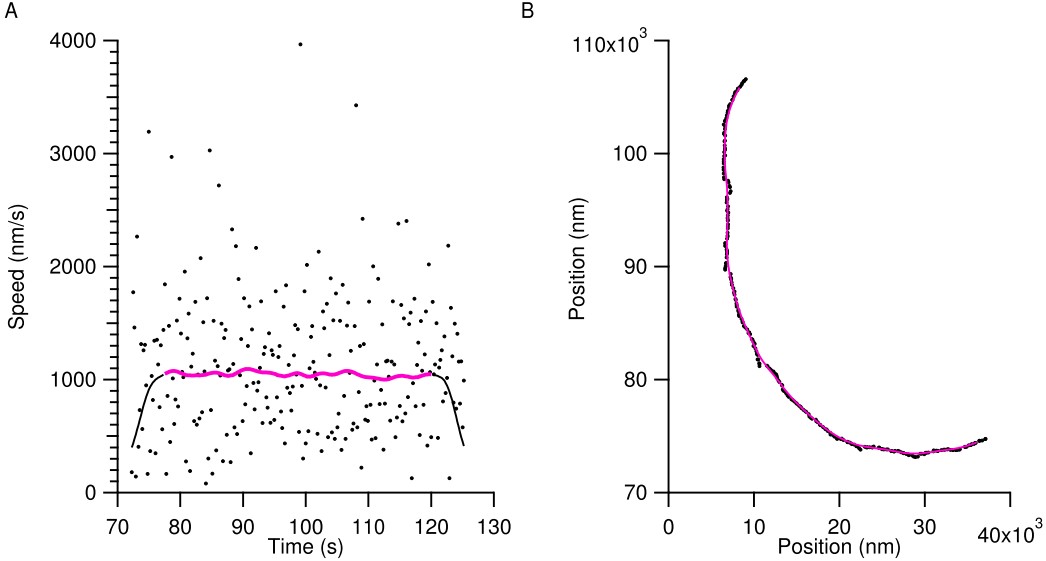

**Figure 1 Characteristic tracking data from a microtubule in the light water buffer.** The markers in (A) show the instantaneous speed of the raw tracking data for the trailing end of a single microtubule and the raw positional data is shown in (B) as markers. The wide scatter of the markers in (A) is due to transverse motion of the microtubule ends to the trajectory and obscures the gliding speed. To account for this, raw positional time series data were smoothed as described in the text, resulting in the solid magenta line in (B). The solid black line in (A) shows the instantaneous speed of the smoothed positional data. To account for edge-effects of smoothing, 5 s were removed from the beginning and end of the smoothed speed, resulting in the final speed dataset used for analysis, shown as the solid magenta line in (A).

isotopes on microtubule speeds, as there are a greater number of lower speed values as compared to the light water buffer.

Figure 3 shows the results of a light water buffer assay where a simple histogram of smoothed concatenated speed values was generated using a 30 nm/s bin width and the corresponding KDE analysis shown in the solid red line. Using the KDE analysis captures all the features of the histogram without the need to adjust bin sizes. It also allowed us to locate the most probable speed in which the microtubules within an assay were traveling at with a higher degree of accuracy. The large kernel width also reduced the sensitivity to speed changes due to the number of kinesin molecules pulling on a microtubule seen by *Gagliano et al. (2010)*.

## RESULTS AND DISCUSSION

### Results

Figure 4 shows speed variations using increasing amounts of heavy-hydrogen water (triangles) and heavy-oxygen water (circles) in the motility solution. Each data point represents three independent measurements where the error bars are the standard error of the mean. Several hundred microtubules were tracked for each data point at a constant temperature of $33.1 \pm 0.1\,°C$. The most likely speed using 100% light water was found to be $1016 \pm 11$ nm/s for the heavy-hydrogen water experiment and $1007 \pm 4$ nm/s for the heavy-oxygen water experiment. Increasing the amount of either the heavy-hydrogen
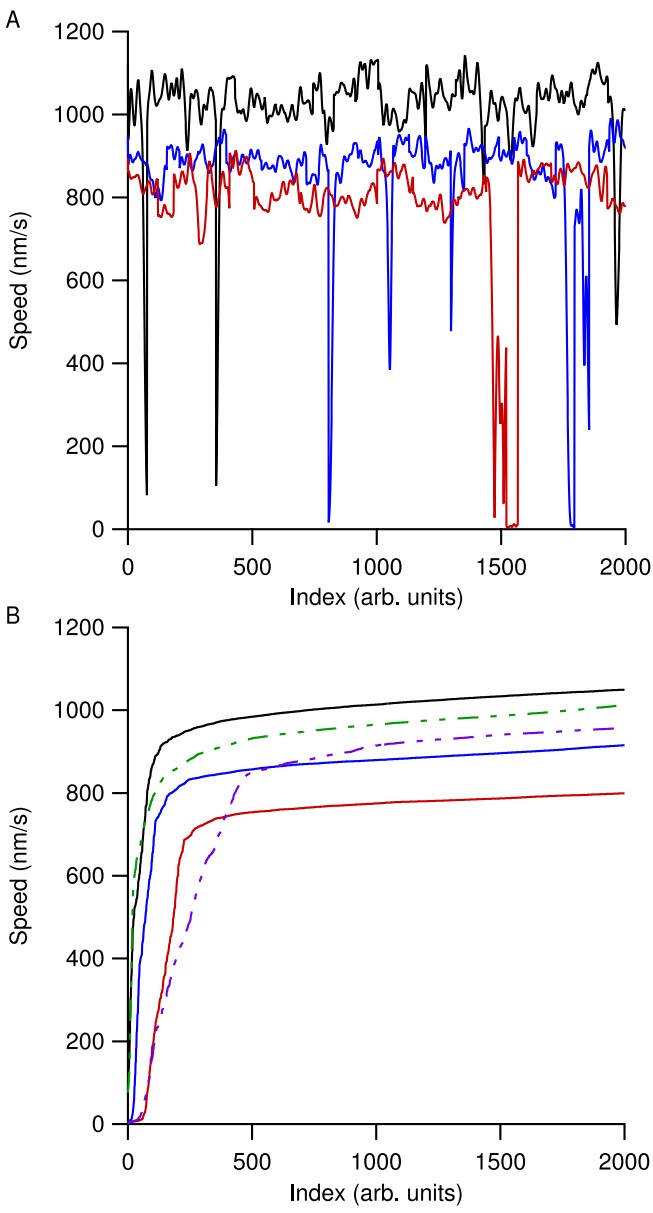

**Figure 2 Concatenated smoothed speed values for varying amounts of water isotopes in the motility buffer.** (A) Concatenated speed measurements for microtubules using heavy-hydrogen water. The black line has 0%, blue 50%, and red 90% heavy-hydrogen water in the motility solution. (B) Sorting the speed values in (A) shows the effects of pausing and stalling on microtubules in the various water isotopes. The solid lines used the heavy-hydrogen water and the dashed lines used the heavy-oxygen water. Solid lines have the same values as in (A) and the green dashed line and purple dashed line had 45% and 81% heavy-oxygen water in the motility solution, respectively.

water buffer or the heavy-oxygen water buffer used in the motility solution caused the most likely speed measurement to decrease in a linear fashion. At the highest concentration of heavy-hydrogen water buffer used, the most likely gliding speed was found to be $799 \pm 8$ nm/s while the heavy-oxygen water buffer was found to be $954 \pm 16$ nm/s. The

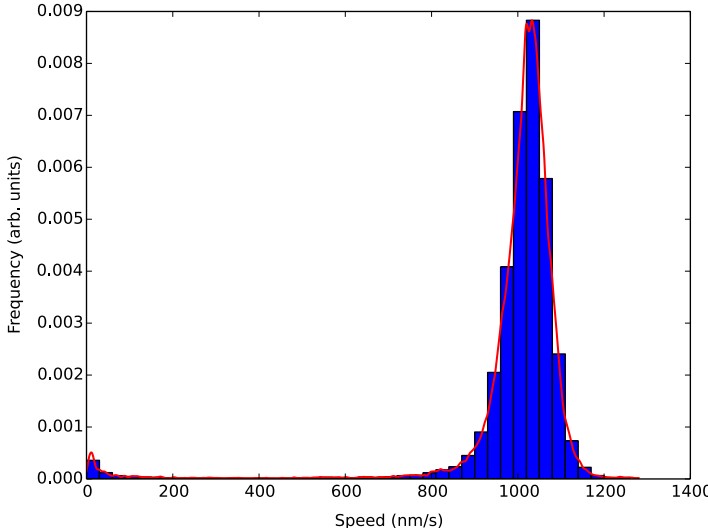

**Figure 3 Comparison of the KDE algorithm with a histogram of smoothed concatenated speed values.** The bins correspond to a simple histogram of smoothed concatenated speed values using a 30 nm/s bin size for a light water buffer assay. The figure corresponds to over 4000 speed points from over 400 different tracked microtubules. The red solid line is a KDE analysis of the smoothed speed values. The KDE algorithm correctly mimics the features of the histogram and allows for greater granularity when determining the most probable speed of microtubules within the assay.

decrease in speed as compared to the light water buffer for the heavy-hydrogen water was approximately 21% while for the heavy-oxygen water it was a decrease of 5%. The deuterium-depleted light water buffer is also represented in this figure. The most probable speed at which the microtubules moved at in the deuterium-depleted light water buffer was found to be $1015 \pm 4$ nm/s.

Figure 5 shows that heavy-hydrogen water does not cause permanent damage to the kinesin-1 surface. Using the resealable flow cell, we initially observed microtubules gliding in a light water motility solution for our standard 15 regions. We then exchanged the light water motility solution with twice the volume of the sample cell with the heavy-hydrogen water motility solution. We then resealed the flow cell with cellophane and observed microtubules in the heavy-hydrogen water buffer. As can be seen in Fig. 5, the average measured speed of microtubules dropped from $987 \pm 2$ nm/s to $791 \pm 1$ nm/s when the light water motility solution was exchanged with the heavy-hydrogen water motility solution. In order to ensure that heavy-hydrogen water did not damage the kinesin-1 surface irreversibly, another fluid exchange from the heavy-hydrogen water motility solution back to the light water motility solution was done. Measured microtubule gliding speed values recovered back to $960 \pm 2$ nm/s thus indicating that heavy-hydrogen water does not damage the kinesin surface.

Comparing the methods of sealing the flow cell with cellophane and nail polish shows that using nail polish, as the sealant, does not interfere with motility. Nail polish contains organic solvents and other components that can leach into the motility solution and cause

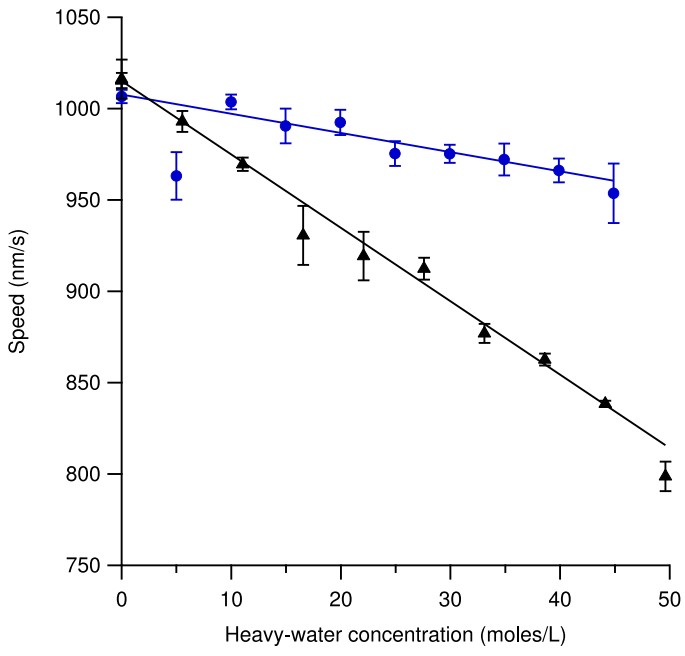

**Figure 4 Microtubule gliding speed at 33.1 ± 0.1 °C versus concentration of water isotope in solution.** Markers represent the mean of observations from three independent samples. Triangles are the heavy-hydrogen water and circles are the heavy-oxygen water. Each of the three observations was an estimate of the most likely gliding speed of hundreds of microtubules in the sample after temperature equilibrium was reached (see text). Error Bars represent the standard error of the mean of the three observations. The solid line is a linear fit to guide the eye and does not represent a fit of a theoretical model. The steady decrease in speed (as opposed to a roll-off at intermediate concentration) indicates that the added deuterium affects a rate-limiting step for microtubule gliding speed on kinesin-1 surfaces. Not shown in the graph is the gliding speed in deuterium-depleted water (1015 ± 4 nm/s), which was indistinguishable from light water (1016 ± 11 nm/s).

damage to the assay. Measured speed values using both nail polish and cellophane as the sealant show that stable speeds can be measured and that nail polish does not damage motility observations over the time span of our experiments, 30 min for each slide.

## Discussion

Adding light water buffer to the heavy-hydrogen water buffer to measure speed variations due to $^2H_2O$ concentrations in the motility solution inevitably caused variations in the final pH of the solution. This was due to the fact that the light water buffer was pH-ed to a value of 6.89 while the pH of the heavy-hydrogen water buffer was 7.30. In order for the two solutions to have the same effective pH, the heavy-hydrogen water buffer should have been pH-ed to 6.48, which would have made the pD 6.89. *Böhm et al. (2000)* showed that pH does affect the gliding speed of microtubules. However, they showed that speed increases with increasing pH and showed that pH values between 6.89 and 7.71 increased measured gliding speeds by approximately 50 nm/s. Figure 4 shows that the speed difference from a solution containing all light water buffer (pH of 6.89) and one containing nearly all heavy-hydrogen water buffer (pH of ∼7.70) decreased

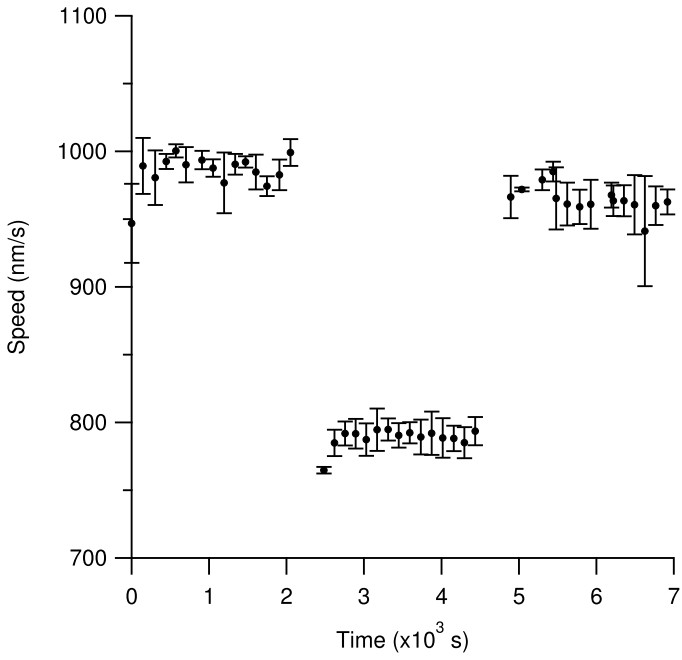

**Figure 5 Determination of any deleterious effects of heavy-hydrogen water on the gliding motility assay.** Time is measured relative to the flow-in of the first motility buffer. Each solid circle represents the most likely gliding speed from the microtubules in one region of observation over a two-minute period. Each set of measurements reveals the initial time necessary for the sample temperature to obtain equilibrium with the objective temperature. The first cluster of data represents observations with the light water buffer. An average speed of $987 \pm 2$ nm/s was measured for the last 10 data points. For all stated values, the first five data points were excluded due to the slide reaching an equilibrium temperature with the objective. The middle set of data represents observations after replacing the light water buffer with twice the volume of the flow cell with the heavy-hydrogen buffer. The measured average speed was $791 \pm 1$ nm/s. The final set of data represents observations after re-filling the flow cell with twice the sample volume of light water buffer. The average measured speed was $960 \pm 2$ nm/s. The return to nearly the same speed as the first data set indicates that heavy water does not cause severe permanent changes to the kinesin surfaces. Error bars are the standard deviation of the average for that ROI.

measured speed values by approximately 200 nm/s. Due to the fact that our measured speed values are of greater magnitude and in the opposite direction expected from pH increase indicates that the effect we see is due primarily to isotope exchange and not to pH variation. Obtaining the pH value of 7.30 also required the addition of more NaOH to solution. Doing so also increased the ionic strength of the solution, which Böhm showed should increase the gliding speed, which is again opposite to the effect we observed (*Böhm et al., 2000*).

In light water, there exists approximately 17 mM of deuterium in solution, which exist mostly as HOD molecules (*Somlyai et al., 1993*). In order to determine if this small amount of deuterium in solution affected the gliding speed of microtubules, a motility solution that was made of deuterium-depleted light water was used. Every possible effort was made to ensure that the trace amounts of deuterium in this motility solution were as low as possible. We found that the gliding speed of microtubules using a deuterium-depleted light water buffer was $1015 \pm 4$ nm/s. This value agrees with the intersection of the fit line in Fig. 4

and was indistinguishable from light water speed measurements. We were unable to detect effects of deuterium-depletion in the gliding motility assay, as expected.

There exists a measurable drop in speed values using the resealable flow cell between the initial observations using the light water motility solution and the final observations using the light water motility solution (see Fig. 5). This drop in measured speed values was attributed to fluid exchanges in the resealable flow cell. Our initial experiments to determine the viability of a resealable flow cell used light water exclusively where we exchanged the fluid in the cell multiple times, see Text S1. Over successive fluid exchanges, the measured speed values decreased every time the fluid was exchanged and the resulting microtubule movement became untrackable after five fluid exchanges. The decrease in speed could be a result of kinesin and passivation being removed from the surface during successive fluid exchanges.

We observed that both heavy-hydrogen water ($^2H_2O$) and heavy-oxygen water ($H_2{}^{18}O$) decrease the gliding speed of microtubules in a linear fashion. A similar decrease was observed for myosin and actin when substituting deuterium oxide for light water (*Chaen et al., 2003*). This indicates that both water isotopes affect a rate-limiting step for transport in the kinesin gliding assay. Kinesin bulk assays and single-motor assays suggest that the rate-limiting step is inorganic phosphate release and/or kinesin rear head release (*Ma & Taylor, 1997*; *Moyer, Gilbert & Johnson, 1998*; *Guydosh & Block, 2009*; *Muthukrishnan et al., 2009*). The rate-limiting step in the gliding assay has not been explored as thoroughly, and it may be affected by surface-effects not observed in bulk or single-motor assays. We note that the proportional decrease in gliding speed is similar to the proportional reported increase in viscosity of the two pure isotopic solutions (*Hardy & Cottington, 1949*; *Kudish, Wolf & Steckel, 1972*). We did not attempt to measure the viscosity of any of our fractional mixtures.

Increased microviscosity could be the cause of our observed decrease in gliding speed, which could possibly slow the rate of inorganic phosphate release (*Blacklow et al., 1988*; *Benner, 1989*). Microviscosity is also known as microscopic viscosity, which is the viscosity a single protein would be affected by due to the medium immediately surrounding it (*Nemet, Shabtai & Cronin-Golomb, 2002*). If the effect we observed is due to microviscosity, then using water isotopes as a microviscosity probe could probe the effects of diffusion-controlled, rate-limiting steps in kinesin and other enzymes. Another method to perturb the viscosity of the system—and ultimately the osmotic pressure of the system—would be to use osmolytes such as betaine, sucrose, or polyethylene glycol. Changing the osmotic pressure of the gliding motility assay does affect microtubule gliding speeds and could also be used as a method for observing rate-limiting steps (*Böhm et al., 2004*). Thus, water isotopes, particularly $H_2{}^{18}O$, which has less effect on hydrogen bonding than $^2H_2O$, may provide important complementary information for osmotic stress studies.

*Chaen et al. (2001)* discuss the slowing of actomyosin in heavy-hydrogen water to be a result of a higher degree of hydrogen bonding to ADP within its motor domain thus indicating that the use of heavy-hydrogen water in the assay induces a slower hydrolysis of

ATP. Our results show a similar slowing effect of microtubules in the gliding motility assay due to an increase of heavy-hydrogen water and heavy-oxygen water used in the motility solution. The kinetic isotope effect on ATP hydrolysis in myosin using heavy-hydrogen water has been studied by several groups (*Inoue, Fukushima & Tonomura, 1975*; *Hotta & Morales, 1960*), however, the kinetic isotope effect on ATP hydrolysis in heavy-oxygen water has not been studied. Further studies are required to indicate whether or not the slowing effect we observed is due to heavy-oxygen water affecting ATP hydrolysis or if our observations indicate effects due to microviscosity.

## CONCLUSIONS

We have shown that microtubule gliding speeds on kinesin surfaces is systematically slowed by increasing the amount of heavy-hydrogen, $^2$H, or heavy-oxygen, $^{18}$O, water isotopes in the motility solution. The relative effects of heavy-hydrogen water versus heavy-oxygen water are similar to the relative viscosity changes seen in either water type for which microviscosity may provide an explanation for the effect, however, further experimentation and theoretical work is necessary to determine if heavy-oxygen water is causing a slowing of ATP hydrolysis. Water isotopes appear to be an effective experimental knob that can be used to study the effects of water on kinesin activity. Moreover, changing the water isotope (particularly the heavy-oxygen water substitution) could be an ideal method for perturbing the microviscosity of the solvent. Viscosity can be controlled to a certain degree by using different osmotic stress agents that affect viscosity differently. If indeed the effects from heavy-oxygen water are due to microviscosity, the information in this article can be used in combination with osmolyte studies to isolate the effects of osmotic stress.

Finally, some research groups are pursuing usage of molecular motor systems in microdevices (*Bachand et al., 2004*; *Hess et al., 2005*; *Kim et al., 2013*). It is feasible that heavy-hydrogen water is a superior solvent for these systems, due to its stabilizing properties, which include photostabilization (*Sinha et al., 2002*). We observed a 21% speed decrease in microtubule gliding speeds when using 90% heavy-hydrogen water in the motility solution, which may be a demerit against the use of it in microdevices due to its effect on decreasing microtubule speeds. However, the time to catastrophe is greatly increased when microtubules are in heavy-hydrogen water and could outweigh the 21% decrease in speed due to the solvent if the microtubules were to last for several hours past when they would normally depolymerize. The use of molecular motors in microdevices adds a further need for understanding the behavior of kinesin and microtubules in differing water isotopes.

## ACKNOWLEDGEMENTS

We would like to thank Dr. Haiqing Liu for supplying kinesin, and Dr. Susan Atlas for technical discussions.

**Peer**J

### Funding
This study was funded by a DTRA grant: HDTRA-1-09-1-0018 and an INCBN grant from the NSF: DGE-0549500. The funders had no role in study design, data collection and analysis, decision to publish, or preparation of the manuscript.

### Grant Disclosures
The following grant information was disclosed by the authors:
Defense Threat Reduction Agency: HDTRA-1-09-1-0018.
Integrating Nanotechnology with Cell Biology and Neuroscience IGERT: DGE-0549500.

### Competing Interests
The authors declare that there were no competing financial, professional or personal interests that might have influenced the performance or presentation of the work described in this manuscript. LJ Herskowitz is currently employed at Fann Instrument Company and no competing interests: financial, non-financial, professional, or personal influenced the experiments.

### Author Contributions
- Andy Maloney conceived and designed the experiments, performed the experiments, analyzed the data, contributed reagents/materials/analysis tools, wrote the paper, prepared figures and/or tables, reviewed drafts of the paper.
- Lawrence J. Herskowitz analyzed the data, contributed reagents/materials/analysis tools, wrote the paper, reviewed drafts of the paper.
- Steven J. Koch conceived and designed the experiments, analyzed the data, contributed reagents/materials/analysis tools, wrote the paper, reviewed drafts of the paper.

### Data Deposition
The following information was supplied regarding the deposition of related data:
Figshare: http://dx.doi.org/10.6084/m9.figshare.963548.

### Supplemental Information
Supplemental information for this article can be found online at http://dx.doi.org/10.7717/peerj.284.

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
