# Peer review of "Effect of 2H and 18O water isotopes in kinesin-1 gliding assay"

_PeerJ, doi:10.7717/peerj.284_

## Round 0.1 · original submission · Minor Revisions

In your revised manuscript, please, address the reviewers comments.

Reviewer 1 ·

Basic reporting

The authors report the effect of isotopes of hydrogen or oxygen in water used as a media on motility of a kinesin-microtubule system. They observed a linear decrease of the gliding velocity of microtubules driven by kinesin using hydrolysis reaction of ATP upon increasing the ratio of D2O or H218O against light water. They insist that the effect could be explained by the change of microviscosity.
The topic is interesting and the experiment is well done. However, the observed isotope effect is possibly explained by some other factors. One possibility is the hydrolysis rate of ATP. It is well known that heavy water induces a slower solvolysis. If authors think that viscosity effect is more decisive, they need to introduce all of the possibility and explain the reasons for thinking viscosity effect as the most provable one.
In the conclusion section they mention about the importance of the heavy water effect in the application of motor proteins to microdevices. The stabilizing properties of heavy water are not a clear concept. Slower velocity of motor proteins can be a demerit in such devices. The authors should explain merits of usage of heavy water in the device in detail.

Experimental design

No comments

Validity of the findings

No comments

Additional comments

No comments

·

Basic reporting

The manuscript by Andy Maloney and coworkers reports on kinesin-1 gliding assays in the presence of heavy-oxygen and heavy-hydrogen water. The authors observed a significant, linear decrease in microtubule gliding speeds as a function of increasing water isotope content. The data appears to be technically correct and sound. However, the manuscript would benefit from a few minor additions and changes which should be addressed before publication.

1. Please remove claims of novelty from the first paragraph.

2. It would be helpful to show some of the original data, i.e. representative traces of gliding position and speed versus time for the different buffer conditions. In this manner, other effects of the water isotopes, e.g. on pause frequency and duration may show up. Also, to show the effect of the various filters, the raw data and filtered traces should be shown. To judge the effectiveness of the KDE algorithm, a speed histogram of speeds determined by linear fits to segments of the raw data traces should be shown. In such histogram, the KDE value and the mean should be indicated when the same data is analyzed by the KDE algorithm.

3. Figure 1 and 2 should be combined in one figure with two panels to avoid a lot of duplication in text.

4. The text should be double-checked that 'heavy-hydrogen water' and 'heavy-oxygen water' are always referred to as such and not just by 'heavy-hydrogen' (e.g. in line 223 it should read 'heavy-hydrogen water buffer' or in the figure caption 1 & 2 in the abbreviations - there are more incidences).

5. Please define microviscosity for the non-specialist.

6. Delete sentences that refer to future work (line 309 & 324).

7. In the figure caption of Fig. 1 & 2, it should read: 'The solid line is a linear fit to guide the eye and does not...'

8. How were the averages calculated in Fig. 3? Were all data shown included in the average? Error bars are shown in the figure. Please delete the wrong statement in the caption.

Experimental design

No comments

Validity of the findings

No comments - see basic reporting.

---

## Round 0.2 · accepted · Accept

Your manuscript has been accepted, there are no comments.